# Effectiveness and Challenges in Local Self-Governance: Multifunctional Autonomy in Japan

**DOI:** 10.3390/ijerph18020574

**Published:** 2021-01-12

**Authors:** Ryuichi Ohta, Yoshinori Ryu, Daisuke Kataoka, Chiaki Sano

**Affiliations:** 1Community Care, Unnan City Hospital, Unnan City 699-1221, Shimane, Japan; yoshiyoshiryuryu.hpydys@gmail.com; 2Unnan Public Health Center, Unnan City 699-1311, Shimane, Japan; kataoka-daisuke@pref.shimane.lg.jp; 3Department of Community Medicine Management, Faculty of Medicine, Shimane University, Izumo 693-8501, Shimane, Japan; sanochi@med.shimane-u.ac.jp

**Keywords:** aging, community work, local self-government, multifunctional autonomy, rural, empowerment

## Abstract

Community organizing with government support, termed local self-governance (LSG), is a form of policy decentralization for community wellbeing through solutions tailored to local issues. One form of LSG is multifunctional autonomy, in which citizens can comprehensively manage their communities with government support. This study clarified the effect of multifunctional autonomy on healthy life expectancy by assessing related advantages and challenges in rural Japanese communities, using a mixed-methods approach. Disability-free life expectancy from 65 years (DFLE-65) was assessed to compare healthy life expectancies between two rural Japanese cities (with/without multifunctional autonomy). Comparisons revealed better DFLE-65 only among older men in a city with multifunctional autonomy. A cost-effectiveness analysis investigated the relationship between the budget and DFLE-65 change using questionnaire data. Cost-effectiveness analysis of multifunctional autonomy indicated 61,147 yen/DFLE-65. Thematic analysis revealed that multifunctional autonomy created new roles for older men, improving community relationships. However, sustainable multifunctional autonomy in LSG communities may be hindered by a generally aging society, generation gap, and lack of mutual understanding between rural communities and local governments. To ensure the sustainability of multifunctional autonomy, collaborations between local communities and governments and among various generations are critical.

## 1. Introduction

As communities worldwide become more diverse, decentralized governance becomes increasingly essential to ensure effective solutions to community problems [1]. Governmental administrative decisions can affect the specific contents of social work with respect to resource allocation and funding [2]. This is important because each community has unique social problems that require authentic organizational approaches [3]. Decentralized governments with the support of the central government can consider specific cultural contexts and population backgrounds [4], and support people by respecting their perspectives [5]. To effectively solve community problems, community organizing is essential, in which community resources are used effectively and citizens are also motivated to help improve local situations [6]. Government-supported community organizing is referred to as local self-government (LSG), which is a form of decentralization [7]. LSG specifically refers to the self-directed efforts of local people when dealing with community conditions. This arrangement may thus effectively improve community wellbeing through the establishment of solutions that are tailored to local problems and circumstances [8,9].

LSG is known to improve relationships between individuals living within the community. Based on a study from Hungary, despite financial problems of the LSG, each LSG led their community to organize better health and living conditions for them [9]. In India, an LSG was applied in an underdeveloped region to promote the enrichment of human relationships, economic improvement, and citizen’s involvement in community organizing [10]. Furthermore, in China, LSG was established for driving community organizing, which enhanced social capital in communities [11]. LSG enables residents to comprehensively consider situations within their own communities, empowering them to engage in problem-solving [10,11,12]. As potential solutions may directly affect their daily lives, there is increased motivation for residents to remain active in LSG [13]. Previous studies have also shown that LSG may involve various types of people from different backgrounds within the same community, thus increasing their sense of worth when compared to those who do not participate [13]. In conjunction with adequate local government support, effective LSG thus leads to financially and socially sustainable activities [7]. In this regard, LSG may also result in better community health outcomes, especially in rural areas with limited resources [14].

The process of developing LSG depends on cultural and national contexts due to localized variations in communication and collaboration methods [15]. For instance, collectivism affects how people think in many Asian contexts, where important decisions are made communally. This is largely an extension of conservatism and high-context culture [16], and may often be very popular in rural areas [17,18]. In Japan, as aging societies have been advanced, the communities involve various aged people with various ideas, in which LSG can facilitate different people’s ideas. In this regard, multiple processes related to the development of LSG should be clarified [16]; however, few studies have examined LSG development or its level of effectiveness in rural Japanese contexts. Moreover, LSG can improve local people’s quality of life, which can contribute to better life expectancy, but issues directly involving life expectancy have been overlooked in research to date. Thus, our first hypothesis was that LSG can enhance the effectiveness of governance and improve people’s quality of life expectancy in Asian contexts and rural settings. The clarification of the effectiveness of LSG for citizens’ health can drive the application of LSG in other rural contexts. Our second hypothesis was that the application of LSG could clarify specific advantages and challenges for Japanese community members. Japan has the most rapidly aging society in the world, so the effective application of LSG in other country’s communities can be based on this inquiry for the future. This study firstly clarified the effects of LSG on healthy life expectancy by the analysis of a questionnaire about LSG provision and the change of citizens’ quality of life expectancy. Secondly, this study delineated the advantages and challenges related to LSG development in rural Japanese communities by thematic analysis.

## 2. Materials and Methods

### 2.1. Setting

#### 2.1.1. Unnan City

This study was conducted in Unnan City, Shimane, Japan. Unnan City is in the eastern part of Shimane, which is located in the southwest part of Japan, and consists of six districts: Daito, Kisuki, Kamo, Mitoya, Kakeya, and Yoshida (Figure 1). Its total land area measures 553.1 km^2^, which accounts for 8.3% of Shimane Prefecture, most of which is covered by forest. A survey conducted in 2017 revealed that the total population of Unnan City was 38,882 (18,720 men and 20,162 women), with 37.82% being over 65 years of age [19]. Unnan City was the first city that applied LSG to communities in Japan.

#### 2.1.2. Multifunctional Autonomy as One Form of LSG

In Japan, rural communities are applying multifunctional autonomy as one form of LSG. Rural Japanese cities are divided into several communities based on the decisions of each local government, taking land area and population into account; each community is thus allowed to conduct various local activities designed to solve specific social problems. That is, each community is subject to multifunctional autonomy [13]. In this regard, activities are financially supported and monitored by their respective cities. Each community also contains an organization that manages unique activities and engages in social problem-solving; these organizations may employ local people as managers. Community members then relay information about their difficulties and social problems to the organization, which helps develop solutions by creating an environment in which community members can interactively discuss local issues [13]. Each multifunctional autonomy group is funded by each local government and decides its activities independently. Their activities are assessed, and their funding levels decided by local government based on the previous year’s multifunctional autonomy activities.

#### 2.1.3. Multifunctional Autonomy in Unnan City

There are 30 autonomous community organizations in Unnan City, each of which has various functions for managing their respective social issues such as social isolation, accessibility to medical care, and succession of traditional activities. Each district has at least one autonomous community organization: Datio has eight, Kaomo has one, Kisuki has eight, Mitoya has five, Kakeya has five, and Yoshida has three. Each community was separated because of mountainous areas (Figure 1). The average population of the community organizations was 1350 persons (range 148 to 6028). The average population density was 10 to 925 persons/km^2^ (range 10 to 925). Each community had different groups that had specific functions (e.g., community organizing, healthcare, and continual education). Each autonomous community organization contains a director, sub-directors, and clerks. Multifunctional autonomy in Unnan city consists of three main categories of operation: community organizing, healthcare, and social/environmental development. Community organizing refers to citizen empowerment, the effective utilization of local resources, and solving community problems through the efforts of local actors. Healthcare involves primary prevention and care improvements, effective healthcare-seeking behaviors, welfare enrichment, and community healthcare satisfaction. Finally, social/environmental development entails the construction of environments and societies that are friendly to citizens while creating a communal sense that life is worth living through education, historical knowledge, and cultural preservation.

### 2.2. Measurements

#### 2.2.1. Framework for the Assessment of Multifunctional Autonomy: RE-AIM Framework

The relevant assessment framework for understanding the conditions and provisions of intervention is “reach, effectiveness, attainment, implementation, and maintenance” (RE-AIM). Previous studies have shown that RE-AIM can effectively promote community organizing by comprehensively engaging all citizens [20]. In this study, to assess the dimensions of reach, attainment, implementation, and maintenance, we used serial cross-sectional investigation based on the annual city questionnaire. To assess effectiveness, we measured healthy life expectancy and cost-effectiveness. To inquire in depth into the processes by which multifunctional autonomy is provided, the content of community forums and annual conferences was used for thematic analysis.

#### 2.2.2. Serial Cross-Sectional Investigation Based on the Annual City Questionnaire

To assess the effectiveness of multifunctional autonomy provision based on citizens’ perceptions, this study obtained data from an annual questionnaire used to assess individual community conditions in Unnan City. The questionnaire was distributed annually to 2000 randomly chosen citizens aged 20 years or older (annual response rates range from 38.5 to 56.4%). Components cover the three areas: community organizing, healthcare, and social and environmental development. The community organizing component contains items on interests, participation, collectiveness, and effectiveness, while the healthcare component contains items on satisfaction with medicine, primary care physician usage rates, efforts to prevent health conditions, regular exercise, and participation in welfare activities. The social and environmental development component includes items on community safety, satisfaction with city living, the sense that life is worth living, and social interaction. All items are rated on a 4-point Likert scale ranging from strongly agree to strongly disagree.

#### 2.2.3. Data on Healthy Life Expectancy, LSG Costs, and City Backgrounds

This study assessed healthy life expectancy as disability-free life expectancy from 65 years of age (DFLE-65). Sullivan’s method was used to calculate the duration of DFLE-65 based on data from the Japanese long-term care insurance system [21]. In this regard, the Japanese government assesses the care needs of individuals over 40 years of age who have disabilities due to disease. These persons with disability are categorized as either having no care needs or are assigned one of five care-need levels (1 is lowest, while 5 is highest). More specifically, DFLE-65 was calculated on the basis of the number of disabled persons aged 40–60 years with care levels of 2 or higher. The DFLE-65 for each year was calculated according to a five-year average, that is, including the two previous and two following years. We assessed changes in DFLE-65 in Unnan City based on comparisons with another city in the same prefecture (City A). City A was anonymized because of the request of the city hall. Importantly, City A did not implement LSG and was not in contact with Unnan City. City A was chosen for comparison because it shared the same background as Unnan City in terms of culture, socioeconomic status, and population aged over 65 years, but neither city geographically affected the other. DFLE-65 was calculated for both cities based on national and city-level data on population size, average age, sex, proportion living alone, and long-term care insurance usage. The annual budget for LSG was collected from Unnan City and analyzed for cost-effectiveness.

#### 2.2.4. Community Forums and Annual Conferences

The city hall organizes annual community forums to discuss the conditions of local autonomous community organizations. This helps communities determine the advantages and disadvantages of each autonomous organization, which then aids in developing new and revised city planning efforts. Conference participants include the directors and staff members of each autonomous community organization, with city hall clerks facilitating each meeting. The staff members of each autonomous community organization have 20 min to present their activities and discuss the successes and difficulties they encountered. Discussions are held after each presentation. In each community forum, three autonomous community organizations make presentations. In this manner, a total of 30 forums have been held over a 10-year period. All discussions are recorded and transcribed verbatim, and then stored for 10 years. The city hall also holds an annual conference with each autonomous community organization to better understand the conditions and challenges related to multifunctional autonomy. Clerks visit the community center of each organization, where they hold discussions with directors and organizing members for 1–2 h. Again, all discussions are recorded and transcribed verbatim.

### 2.3. Analysis

We conducted a quantitative analysis of questionnaire data by dichotomizing item answers as either agree (strongly agree + agree) or disagree (disagree + strongly disagree). Data trends were graphically described and analyzed using chi-squared tests and Kruskal–Wallis tests, while Student’s t-test was used to analyze parametric data. Statistical significance was set to α = 0.05. Cost-effectiveness analysis was performed by calculating the total amount needed for 1 DFLE-65 per person over a 7-year period. The total cost of LSG per person was divided by the increase in DFLE-65 over 7 years when compared to City A.

For the qualitative analysis, a thematic analysis approach was used to clarify the effectiveness and challenges related to multifunctional autonomy [22]. The first and second authors carefully read transcriptions of community forum discussions and annual conferences, to gain familiarity with the content. Initial coding was then conducted using the content of each discussion [22]. After reading each discussion transcript, the first and second authors discussed their understanding of multifunctional autonomy. Next, the first author performed primary initial coding and formulated a codebook containing relevant definitions and examples. After each initial coding of the discussions, the codebook was shared with the second author, who also read the transcripts and coded a subset based on the codebook. The first and second authors frequently met to discuss their coding processes. As such, coding was refined via constant comparison, merging, or deleting codes until consensus was reached. The first and second authors then reviewed their open coding and identified overarching themes and subthemes to describe the effectiveness and challenges of multifunctional autonomy [22]. For all disagreements, the authors returned to the transcripts and confirmed their understanding until reaching an agreement. The final themes and concepts were discussed among all authors until mutual agreement was achieved.

### 2.4. Ethical Considerations

Participants were informed that all questionnaire data would solely be used for research purposes. Furthermore, all questionnaire data were anonymized. The questionnaire instructions also contained information on the research aims, type of data to be disclosed, and how personal information would be protected. The instructions clearly stated that study consent was given by answering the questionnaire. In this regard, informed consent was also obtained before beginning all interviews and focus groups. We received permission from the city hall to use the aforementioned contents of previous discussions. This study was approved by the Unnan City Hospital Clinical Ethics Committee (ethic code: 20200010).

## 3. Results

### 3.1. The Effectiveness of Implementing Multifunctional Autonomy

A serial cross-sectional investigation into the provisions of multifunctional autonomy generally revealed that most components of the category regarding the effectiveness of related activities remained at a high rate. In the category of community organizing, the components of interest and participation were greater than 70% through the duration of the study. In the category of living environment, all the components were greater than 60% through the duration of the study. In the category of healthcare, the components of satisfaction, PCP, and health maintenance were greater than 70% through the duration of the study, although the component of exercise was not improved and maintained below 40%. Regarding welfare activities, the rate of community engagement gradually improved from 23–36.4% (Table 1).

### 3.2. Cost-Effectiveness Analysis and Changes in DFLE-65 between Unnan City and City A

In City A, the population of individuals aged 65 years and above was 52,368 (28.3%) in 2005, 50,015 (31.0%) in 2010, and 47,718 (35.1%) in 2015. A comparison between Unnan City and City A revealed persistent differences in DFLE-65 length, although this did not widen from 2009 to 2013 (both men and women). However, the difference in DFLE-65 widened beginning in 2014, with the differences widening from 0.17 to 0.73 among men between the two cities (Figure 2); this did not occur among women (Table 2). The difference in DFLE-65 consistently ranged from 0.23 to 0.28 among women.

The cost-effectiveness of LSG was calculated with the average SSG budget, and the average population and change in the DFLE-65. The average LSG budget for 7 years was 269,645,000 yen (SD = 19,876.59). The average population was 42,384 (1181.9). Based on the cost-effectiveness analysis, 61,147 yen/person was needed for an increase of 1 DFLE-65/person among men, and 159,419 yen/person was needed for an increase of 1 DFLE-65/person among women.

### 3.3. The Effectiveness and Challenges of Multifunctional Autonomy through Qualitative Analysis of Interviews and Discussion Contents

Based on the thematic analysis, two themes appeared: the effectiveness and challenges. The theme of the effectiveness consists of three concepts: building new roles for citizens, deep understanding of communities, and effective collaboration among citizens. The theme of challenges consists of need for transformation, a generational gap affects the community’s future, and lack of mutual understanding between governments and autonomous organizations (Table 3).

### 3.4. The Effectiveness

#### 3.4.1. Building New Roles for Citizens

In the context of traditional multifunctional autonomy, rural citizens needed to analyze community conditions and manage various events/projects to improve their own lives. Therefore, a variety of citizens took up roles in community management. As many young citizens had daytime jobs, older citizens typically took most organizational positions. One participant stated the following:
“Community organizing is tough because we have to consider history and social conditions. We are retired and have some free time to use for community organizing. Older men have time to devote to community organizing because of their retirement. Most may work in agriculture, but working time can be limited… Although some tasks in community organizing are onerous, I feel a sense of accomplishment in this new activity and want to continue it in sustainable forms.”

Although community members experienced related difficulties, they were also motivated to engage in community development because it helped them gain skills after retirement. This was especially common among older men. By reflecting on older people’s present conditions, they realized the importance of LSG for older people’s health. Another participant stated the following:
“Role making can be effective, especially among older people. The present cohort of older people are healthy and active. Although there is a retirement age in a lot of companies and organizations, older people, especially older men, can work and enjoy their lives. Autonomous community organizations can be important for older people to remain active, which can make them healthy mentally and physically.”

#### 3.4.2. Deep Understanding of Communities

Although many rural citizens were anxious about the future of their communities, they did not hold official meetings and were unable to involve all residents. In the context of LSG, multifunctional autonomy thus led various citizens to seek inclusion in future community planning efforts. The participants acknowledged that to prepare for an aging society that is also inclusive, LSG is essential. One participant stated the following:
“We did not know the current conditions of the communities. We may have avoided these considerations, although no official time was allocated to consider such opportunities. In community organizing, we must confront the various severe realities of our communities… Although it may be difficult to confront community realities, the process of considering the future can lead to an understanding of our communities. Besides, the process may help rural citizens realize the advantages and disadvantages of improving community conditions.”

Through the involvement in the activities of LSG, rural citizens gain knowledge of many situations, which provide them with an understanding of their community’s strengths and challenges while engaging in community organizing. This contributes to a deeper overall understanding of the realities of community life. The participants realized that mutual understanding facilitated activities for future communities. Another participant stated the following:
“Thanks to the autonomous community organization and multifunctional autonomy, citizens in each community become interested in events in the communities and in the future and sustainability of communities… During the community organizing discussions, various community members can insist on their opinions about various groups in the community, which can lead to a better understanding of community’s activities and future.”

#### 3.4.3. Effective Collaboration among Citizens

Multifunctional autonomy provides various citizens with opportunities to collaborate during events and projects. This also gives them ample time to have in-depth dialogues about their lives. For example, they can share information about perceived future difficulties and anxieties. Through the dialogues, the rural citizens observed the unity of the communities and saw the positive future of their communities. One participant stated the following:
“Multifunctional autonomy has given us more opportunities to meet members in the same communities than in the past. We are holding various events in our communities with other members through constant discussions… During the events, we can feel deep connections with others and understand community activities. In the discussions and events, we can share our concerns about the future. These perceptions and opportunities can contribute to member motivation and effective collaboration.”

Through collaboration in their communities, residents can better understand one another, while simultaneously gaining a sense of community collectiveness through this process, which could help enhance collaboration. Another participant stated the following:
“Through the participation in community organizing, I can communicate with others with whom I usually do not confer. For example, in planning recreational activities for the elderly, new ideas can appear through discussion by using new resources in communities, which we do not know about originally… By effectively collaborating with each other, even rural communities can create new things for the sustainability of communities.”

### 3.5. Challenges

#### 3.5.1. Need for Transformation

Rural communities are confronted with a situation in which younger generations are moving to other locations. As society continues to age, so do the individuals engaged in LSG. This makes it difficult to find new applicants who will replace them in their roles. The community members struggled with the transformation of their LSG. One participant stated the following:
“The sustainability of local self-governance is essential, but the present conditions may not allow sustainability. The outflow of young generations and our aging society strongly affect rural communities. We are getting old and must transfer our positions to the younger generation. However, thinking about the social conditions in Japan, our community situations will not change. Of course, we should educate the younger generation. But not only that. We have to think about the transformation of local self-governance, such as merging with other communities or scaling down activities.”

Though generational change is essential, the lack of human resources inhibits the overall process. However, sustainable LSG requires continuous education to prepare successive generations, who are expected to transform their communities. The rural community members were anxious about the sustainability of LSG in an aging society. Another participant stated the following:
“Young people may not be interested in community organizing, because many of them work outside the communities during the day. They may think that after retirement, they will work in the communities. However, the present aging speed is rapid and the LSG is sustained by a limited number of older people. Therefore, we have to prepare for their retirement for the sustainability of the communities.”

#### 3.5.2. A Generational Gap Affects the Community’s Future

Multifunctional autonomy is tremendously affected by a generational gap resulting from changing social conditions. That is, younger generations are not motivated to engage in multifunctional autonomy because they have not participated in community organizations. Instead, older residents make most decisions in the context of rural community organizing, thus preventing younger generations from making those decisions. Further, younger generations believe that older generations do not respect their ideas due to conservative rural conditions. They, then, feel that they lack understanding about community organizing, as a whole. One participant stated the following:
“Rural communities are conservative and difficult to change. Although there are young people who are motivated to improve the present conditions of rural communities, they may lose their motivation because of less cooperation from older people. Besides, as the younger generation is not used to participating in community organizing, they cannot participate in rural communities… Rural communities should have educational systems about community organizing, with productive help from the older generation, which can encourage the younger generation to engage in community organizing. The older generation should discuss how to effectively involve the younger generations without their efforts being rejected by the community.”

In contrast, older generations believed that younger generations were too busy and unmotivated to engage in community organizing. The older generation thus experienced difficulty when transferring community roles. One participant stated the following:
“Commonly speaking, the younger generation may not have the motivation to participate in community organizing… As the community is aging, more and more young people should participate in community organizing and take essential roles there. Trying to figure out how to motivate them to participate in community organizing has been difficult.”

#### 3.5.3. Lack of Mutual Understanding between Governments and Autonomous Organizations

After initiating LSG, rural citizens were motivated to manage autonomous organizations and improve community conditions. However, the continuous allocation of city hall tasks to autonomous organizations was an exhausting process, sometimes even extending to the limits of their capacity. One participant stated the following:
“The present conditions of community organizing are not sustainable for the future. In addition to the lack of a workforce and the aging society, the tasks presented to autonomous organizations have continuously increased. The city hall must take more account of the limitations of rural community conditions. We have to think about the balance between community capacities and the burden of the tasks. There should be an increase in the number of city hall-driven jobs related to community organizing.”

Rural community members were then required to consider the difficulties associated with the sustainability of having autonomous organizations proceed to multifunctional autonomy. At that time, they realized the need to establish mutual understanding with local governments concerning their community conditions. Another participant stated the following:
“The possibility of LSG can be high if there are a lot of resources in each community. However, in aging societies, there are few resources, especially human resources. Only financial support from the local government is not enough for the continuity of LSG. Sharing human resources between governments and autonomous organizations can be urged, and we should know each other’s working conditions. The present situation may not be ideal because each side tends to pass jobs to the other.”

## 4. Discussion

This study clarified the effectiveness of the continuous provision of multifunctional autonomy in LSG from the perspective of improving healthy life expectancy, specifically in regard to DFLE-65 among older men. Qualitative analysis showed that such an arrangement can result in new roles for older men, which then improves community relationships. In contrast, several factors may hinder the sustainability of multifunctional autonomy in LSG, including an aging society, generational gaps, and a lack of mutual understanding between rural communities and local governments.

Continuously effective implementation of multifunctional autonomy can be vital in LSG. The comprehensive inclusion of all community residents is essential to improve the lives of citizens in the context of community organizing. Based on the RE-AIM framework [20], serial cross-sectional investigation into the provisions of multifunctional autonomy showed that more than 70% of citizens had interests in and motivation for community organizing. This rate persisted from the beginning of multifunctional autonomy, which can lead to better reach, implementation, and maintenance. Further, this investigation showed that 70% of citizens were satisfied with local living conditions and healthcare, which indicates both effectiveness and attainment. Comprehensively speaking, multifunctional autonomy thus appears to be effective and efficient.

The practical implementation of multifunctional autonomy can improve healthy life expectancy for men living in these communities, as shown by the finding that DFLE-65 increased more rapidly for men in Unnan City when compared to City A. Supporting this theory, the cost-effectiveness analysis found that the total cost per capita for LSG was below £20,000/QALY, the amount set by the National Institute for Health and Care Excellence (NICE), which shows the cost-effectiveness of LSG [23]. Multifunctional autonomy also provides various community roles for retired men, which can help them sustain their sense that life is worth living [24]. In general, men tend to lose a sense of functionality after retirement because they previously dedicated their lives to their jobs [25]. As such, retired men may not have hobbies, the lack of which can increase the risk of depression and dementia [25,26,27]. This study showed that older men benefit by working for their communities, through multifunctional autonomy, which may prevent the deterioration of their mental faculties. In contrast, DFLE-65 gradually increased among women in both Unnan City and City A. Although their roles have been altered due to social change, these residents tend to have continuous roles at work and home, including housekeeping and child rearing; this is especially prevalent in rural areas [28]. Multifunctional autonomy primarily appears to modify conditions for men in those communities. As this was a serial cross-sectional study, future studies should investigate personal changes among these individuals in regard to their mental and physical health.

This study’s qualitative analysis showed that rural citizens, especially men, felt effective when taking new roles, thus motivating them to improve their communities. Nevertheless, local government burdens and the problem of transferring roles to the next generation were critical factors for LSG sustainability. Further, new roles were found to be effective for older men. This is also supported by the quantitative results of a study on DFLE-65 among older men [29]. LSG sustainability requires the consideration of issues concerning the transformation of organizational multifunctional autonomy [30]. As an aging society is inevitable, the solutions for current problems should be approached through collaboration with various internal and external organizations, including private companies and non-profit organizations that are interested in community organizing [17,31]. In this context, the availability of multiple resources and functions can reduce the burdens associated with LSG, such as accessibility and availability of resources for lives [32,33]. This process can successfully involve stakeholders in each community and in the local government, which can increase citizens’ empowered activities for others and improve health conditions in rural communities [34,35,36]. Future studies should investigate the specific qualitative and quantitative operational elements involved in this collaborative process. Furthermore, future research should address differences in the impact of multifunctional autonomy on men and women and other factors influencing this difference.

This study also had limitations. One was the use of representative healthy life-expectancy data for whole populations in each investigated city. These data can show trends but cannot clearly reveal cause-and-effect relationships between multifunctional autonomy and healthy life expectancy at the individual level. Future research should thus investigate personal data, thereby taking specific factors related to individual health conditions into consideration. This study was limited because it made comparisons only on the basis of DFLE-65.

Long-term observations are needed to assess the results and effects of community organizing on neighboring cities. This study compared Unnan City to City A, which was a significant distance from Unnan City, even though it belonged to the same prefecture. In addition, although it was known that City A had not implemented LSG, precise background information was not collected due to limited information from local social resources. Future studies should thus investigate changes in DFLE-65 via clustered randomized research involving different prefectures.

## 5. Conclusions

This study showed the effectiveness of multifunctional autonomy in the LSG context, specifically by investigating healthy life expectancy via the continuous provision of activities, especially among older men. We also found that new community roles contribute to these effects. However, various external and internal organizations should collaborate with local communities and governments in order to overcome difficulties associated with sustaining multifunctional autonomy.

## Figures and Tables

**Figure 1 ijerph-18-00574-f001:**
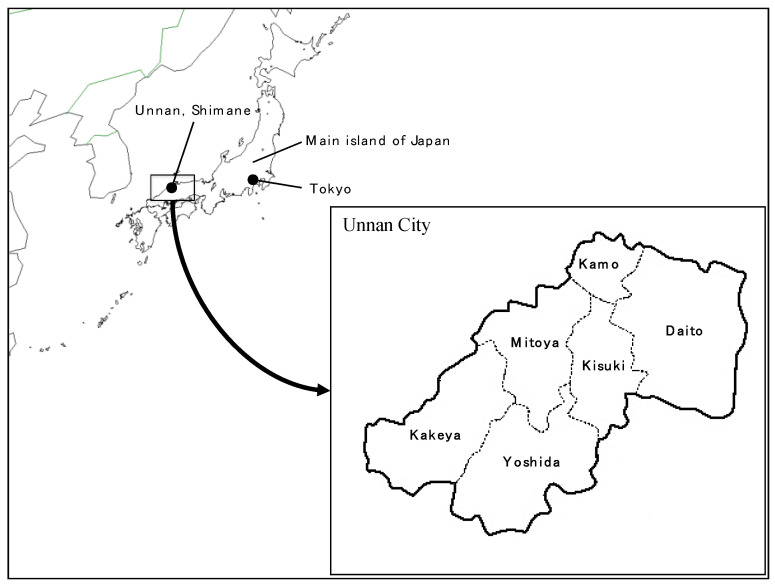
Location of Unnan city and districts.

**Figure 2 ijerph-18-00574-f002:**
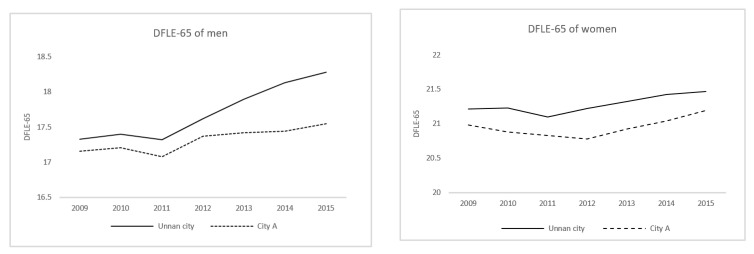
Changes in DFLE-65 for Unnan City and City A.

**Table 1 ijerph-18-00574-t001:** Results of the serial cross-sectional questionnaire regarding multifunctional autonomy.

	2008	2009	2010	2011	2012	2013	2014	2015	2016
Population	44,560	44,019	43,578	42,957	42,279	41,687	41,333	40,850	40,372
% >65 years of age	31.62	31.91	32.1	32.03	32.57	33.67	34.5	35.4	36.09
N (%)	904 (45.2)	890 (44.5)	858 (42.9)	807 (40.7)	869 (43.5)	800 (40.0)	808 (40.4)	768 (38.4)	772 (38.6)
Category, ratio (95% CI)									
Community Organizing									
Interest	0.821 (0.814–0855)	0.773 (0.731–0.811)	0.747 (0.717–0.776)	0.711 (0.679–0.741)	0.725 (0.694–0.754)	0.738 (0.706–0.768)	0.738 (0.706–0.768)	0.738 (0.706–0.769)	0.745 (0.713–0.775)
Participation	0.699 (0.655–0.741)	0.739 (0.696–0.78)	0.693 (0.661–0.724)	0.698 (0.666–0.729)	0.669 (0.636–0.7)	0.716 (0.684–0.747)	0.691 (0.658–0.723)	0.706 (0.672–0.738)	0.729 (0.696–0.76)
Effectiveness	0.571 (0.524–0.617)	0.474 (0.427–0.522)	0.455 (0.421–0.489)	0.473 (0.439–0.507)	0.451 (0.418–0.485)	0.454 (0.419–0.489)	0.562 (0.527–0.596)	0.398 (0.364–0.434)	0.422 (0.387–0.458)
Living Environment									
Safety	0.571 (0.524–0.618)	0.557 (0.51–0.604)	0.569 (0.535–0.602)	0.569 (0.535–0.602)	0.58 (0.546–0.613)	0.6 (0.565–0.634)	0.719 (0.687–0.75)	0.693 (0.659–0.725)	0.685 (0.651–0.718)
Social interaction	0.646 (0.6–0.69)	0.751 (0.708–0.79)	0.666 (0.633–0.697)	0.666 (0.633–0.698)	0.674 (0.642–0.705)	0.649 (0.615–0.682)	0.676 (0.642–0.708)	0.692 (0.658–0.724)	0.705 (0.671–0.737)
SLWL	0.728 (0.684–0.768)	0.746 (0.703–0.786)	0.717 (0.685–0.747)	0.717 (0.685–0.748)	0.709 (0.673–0.734)	0.752 (0.721–0.782)	0.687 (0.654–0.719)	0.68 (0.646–0.713)	0.728 (0.695–0.759)
Comfortability	0.588 (0.542–0.634)	0.571 (0.523–0.617)	0.601 (0.568–0.634)	0.601 (0.568–0.635)	0.565 (0.531–0.598)	0.561 (0.526–0.596)	0.684 (0.651–0.716)	0.671 (0.637–0.704)	0.645 (0.61–0.679)
Health Care									
Satisfaction	0.721 (0.677–0.762)	0.739 (0.696–0.78)	0.718 (0.687–0.748)	0.718 (0.687–0.748)	0.738 (0.707–0.767)	0.682 (0.649–0.715)	0.756 (0.725–0.785)	0.771 (0.74–0.8)	0.834 (0.806–0.86)
PCP	0.761 (0.719–0.8)	0.744 (0.701–0.784)	0.739 (0.708–0.768)	0.739 (0.708–0.769)	0.771 (0.742–0.799)	0.71 (0.677–0.741)	0.731 (0.699–0.762)	0.762 (0.73–0.792)	0.777 (0.746–0.806)
Health maintenance	0.646 (0.6–0.69)	0.665 (0.619–0.709)	0.634 (0.601–0.666)	0.634 (0.601–0.667)	0.641 (0.608–0.673)	0.605 (0.57–0.639)	0.655 (0.621–0.687)	0.649 (0.614–0.683)	0.637 (0.602–0.671)
Exercise	0.374 (0.329–0.42)	0.355 (0.311–0.401)	0.371 (0.338–0.404)	0.351 (0.338–0.404)	0.383 (0.3510–0.416)	0.366 (0.333–0.401)	0.382 (0.349–0.417)	0.378 (0.344–0.414)	0.338 (0.305–0.373)
Welfare activity	0.23 (0.192–0.272)	0.234 (0.195–0.276)	0.249 (0.221–0.28)	0.234 (0.195–0.276)	0.236 (0.208–0.266)	0.354 (0.321–0.388)	0.351 (0.319–0.386)	0.358 (0.324–0.393)	0.364 (0.33–0.399)

Abbreviations: N: number; Interest: Are you interested in community organizing? Participation: Have you participated in community organizing more than once this year? Effectiveness: Do you realize that the community organizing in your community effectively makes your community better? Safety: Do you feel that you have personal safety in your community? Social interaction: Do you usually interact with your neighbors? SLWL: Do you feel a sense that life is worth living (SLWL)? Comfort: Do you feel comfortable in your community life? Satisfaction: Are you presently satisfied with healthcare in your community? PCP: Do you have a primary care physician (PCP)? Health maintenance: Do you do receive regular health check-ups in your community? Welfare activity: Do you usually participate in welfare activities?

**Table 2 ijerph-18-00574-t002:** Disability-free life expectancy from 65 years of age (DFLE-65) differences between Unnan City and City A.

	Men			Women		
Year	Unnan City	City A		Unnan City	City A	
	DFLE-65	DFLE-65	Differences	DFLE-65	DFLE-65	Differences
2009	17.33	17.16	0.17	21.21	20.98	0.23
2010	17.4	17.21	0.19	21.23	20.88	0.35
2011	17.32	17.08	0.24	21.1	20.83	0.27
2012	17.62	17.37	0.25	21.22	20.78	0.44
2013	17.9	17.42	0.48	21.32	20.92	0.40
2014	18.13	17.44	0.69	21.42	21.04	0.38
2015	18.28	17.55	0.73	21.47	21.19	0.28

DFLE = disability-free life expectancy.

**Table 3 ijerph-18-00574-t003:** The results of thematic analysis regarding the effectiveness and challenges of multifunctional autonomy.

Theme	Concepts
Effectiveness	Building new roles for citizens
	Deep understanding of communities
	Effective collaboration among citizens
Challenges	Need for transformation
	A generational gap affects the community’s future
	Lack of mutual understanding between governments and autonomous organizations

## Data Availability

All relevant data sets in this study are described in the manuscript.

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
