# Peer review of "Effectiveness and Challenges in Local Self-Governance: Multifunctional Autonomy in Japan"

_ijerph, 2021, doi:10.3390/ijerph18020574_

Round 1
Reviewer 1 Report
Line 95: “multifunctional autonomies” is unclear. It seems to be a physical office and a forma institution, but autonomy is an abstract term.
Line 138, better a comma than a colon.
Line 140: Why is City A not named?
Line 172: There must be a way to avoid word repetition: “First, the first”
Line 197: “each component of the categories regarding the effectiveness of related activities remained at a high rate”. Not all variables in Table 1 show high rates. Can this be better explained?
Line 203: This question is unclear. Better to try another translation: “Do you realize the effectiveness of community organizing in your community?”
Lines 213-214 What metric are you using for significance? This must be included in Table 2.
Line 216: The findings about women may contradict what it says in the abstract. “Comparisons
revealed better DFLE-65 in a city with multifunctional autonomy, especially among older men.” It may not be “especially”, but “only” men which show a difference. This will depend on statistical tests.
Section 3.2 has only one paragraph. The results in Table 2 need further explanation here and discussion with other studies.
Section 3.3 needs at least a paragraph explaining the subsections
Section 3.3.1: Only one interview sample? Better to find more.
Section 3.3.2 The title is unclear because no explanation was made at the beginning of 3.3. Again, only one quote seems poor.
Section 3.3.3 and sections that follow, again, only one quote is not enough.
Maybe it would be better to add a figure, such as a problem/objective tree, that graphically represents how all the subtopics in Section 3.3 come together. Analysis and discussion with the aid of previous research is lacking.
The research design shows many flaws, including quantitative comparativeness between cities and a more profound qualitative analysis. I believe publication of this data will depend on how much qualitative information can be obtained from available records.
Author Response
Manuscript ID: ijerph-1029689
Title: Effectiveness and challenges in local self-governance: multifunctional autonomy in Japan
Point-by-Point Response
Please note that the changes made do not influence the content, conclusions, or framework of the paper. We have not listed below all minor changes made; however, these are highlighted in red font in the revised manuscript.
Review 1
Line 95: “multifunctional autonomies” is unclear. It seems to be a physical office and a forma institution, but autonomy is an abstract term.
Response: We have revised the description of multifunctional autonomies to multifunctional autonomy performed by autonomous community organization in the whole manuscript.
Line 138, better a comma than a colon.
Response: Accordingly, we have revised the colon to a comma in line 138.
Line 140: Why is City A not named?
Response: We have revised the description of City A with the reason for anonymization based on the suggestion of the information source on the line 139 to 140.
Line 172: There must be a way to avoid word repetition: “First, the first”
Response: Accordingly, we have omitted the word “First” line 190.
Line 197: “each component of the categories regarding the effectiveness of related activities remained at a high rate”. Not all variables in Table 1 show high rates. Can this be better explained?
Response: Yes, we have revised the description of the results from the serial cross-sectional questionnaire regarding each component of three categories on lines 215 to 223 accordingly.
Line 203: This question is unclear. Better to try another translation: “Do you realize the effectiveness of community organizing in your community?”
Response: As per your comment, we have revised the description of the question to “Do you realize that community organizing in your community effectively makes your community better?” on the lines 226 to 227.
Lines 213-214 What metric are you using for significance? This must be included in Table 2.
Response: We have revised the description of Table 2 using the phrase “widening” instead of significant on line 234 to 245 because this process did not include any statistical analysis.
Line 216: The findings about women may contradict what it says in the abstract. “Comparisons revealed better DFLE-65 in a city with multifunctional autonomy, especially among older men.” It may not be “especially”, but “only” men which show a difference. This will depend on statistical tests.
Response: We have revised the Abstract (line 19) and Results (lines 215 to 223).
Section 3.2 has only one paragraph. The results in Table 2 need further explanation here and discussion with other studies.
Response: We have revised the description in the Results section regarding Table 2 and added several sentences in the Discussion.
Section 3.3 needs at least a paragraph explaining the subsections
Response: We have made the necessary changes.
Section 3.3.1: Only one interview sample? Better to find more.
Response: We have added another interview sample to describe the concept in depth.
Section 3.3.2 The title is unclear because no explanation was made at the beginning of 3.3. Again, only one quote seems poor.
Response: We have added an explanation of the themes in section 3.3 as well as another quote.
Section 3.3.3 and sections that follow, again, only one quote is not enough.
Response: We have added explanations and quotes to each theme.
Maybe it would be better to add a figure, such as a problem/objective tree, that graphically represents how all the subtopics in Section 3.3 come together. Analysis and discussion with the aid of previous research is lacking.
Response: We have added a table to categorize the themes and concepts. We also discuss each theme by adding several references.
The research design shows many flaws, including quantitative comparativeness between cities and a more profound qualitative analysis. I believe the publication of this data will depend on how much qualitative information can be obtained from available records.
Response: We have revised the whole manuscript based on the reviewer’s comments.
Reviewer 2 Report
The contribution presents interesting research about the role of Local Self-Government in the definition of healthcare policies in a local community in the city of Unnan. To achieve, elaborate and assess the role, the authors proposed a critical cross-sectional investigation Based on the Annual City Questionnaire, comparing two cases (Unnan and the City A). In their conclusions, they discuss the results of these studies and the potential of these contributions in the definition of a governance strategy.
Despite the interest of the topic and the proposed approach, I would propose to the authors a set of suggestion in order to improve the quality of the article, especially for people that are not familiar with this issue or the Japanese context.
The introduction lacks a clear definition of the final target of the study and a clear definition of the research question. Over the presentation of the methodology and the theoretical context of the precedents, I think the author should point out how these studies could contribute to the definition of sectoral governance and if this potential could be integrated with other fields/sectors. In addition, the introduction should present the overall structure of the article.
Pag. 2/line 45/46: the authors pointed out that “Various forms of LSG are known to improve relationships between individuals living in the same communities”. Is it possible to list these different forms? I think a chart/table that show the different forms and their pro/cons or features could be of interest for a non-expert reader.
Pag. 2/line 63: the authors pointed out that “...few studies have examined LSG development or its level of effectiveness in rural Asian contexts...”. In my opinion, it could be interesting to define the peculiarities of the Japanese context and explain why the LSG aspects of this country could be of interest for a larger audience and the scientific community.
Pag. 2/line 74 and ss.: Is unclear why the authors choose Unnan City as a case study. I appreciated a larger explanation of the reasons and the aspects that made this case interesting on the general portfolio of other cases. This focus makes the research more solid, otherwise, the selection could seem random or not enough motivated
Pag. 3/line 96/97: the authors pointed out that “There are 30 multifunctional autonomies in Unnan City, each of which has various functions for managing their respective social issues”. I think it could be of interest to explain – and describe through data (population affected, area, density, the budget allocated for each autonomy) – the differences among different multifunctional autonomies, to point out how the identified LSG form is peculiar, interesting and fertile in the light developed by the authors.
Authors could have described better the context because I supposed that the effectiveness of the Self-Government depends on several factors, many of them connected to the local conditions and specific constraints. Provided data from annual city questionnaire are interesting, but not enough to understand the role of the LGS implemented in Unnan City as a good/effective practice because authors provide a weak framework for their study. As suggested, once the authors framed the research field and the two cases, the article is interesting and could become more solid.
Author Response
Manuscript ID: ijerph-1029689
Title: Effectiveness and challenges in local self-governance: multifunctional autonomy in Japan
Point-by-Point Response
Please note that the changes made do not influence the content, conclusions, or framework of the paper. We have not listed below all minor changes made; however, these are highlighted in red font in the revised manuscript.
Reviewer 2
The contribution presents interesting research about the role of Local Self-Government in the definition of healthcare policies in a local community in the city of Unnan. To achieve, elaborate and assess the role, the authors proposed a critical cross-sectional investigation Based on the Annual City Questionnaire, comparing two cases (Unnan and the City A). In their conclusions, they discuss the results of these studies and the potential of these contributions in the definition of a governance strategy. Despite the interest of the topic and the proposed approach, I would propose to the authors a set of suggestion in order to improve the quality of the article, especially for people that are not familiar with this issue or the Japanese context.
The introduction lacks a clear definition of the final target of the study and a clear definition of the research question. Over the presentation of the methodology and the theoretical context of the precedents, I think the author should point out how these studies could contribute to the definition of sectoral governance and if this potential could be integrated with other fields/sectors. In addition, the introduction should present the overall structure of the article.
Response: We have revised and added a description of our hypothesis, the significance of the results in other settings, and the overall structure of the article in the background section.
Pag. 2/line 45/46: the authors pointed out that “Various forms of LSG are known to improve relationships between individuals living in the same communities”. Is it possible to list these different forms? I think a chart/table that show the different forms and their pro/cons or features could be of interest for a non-expert reader.
Response: We have revised the description of the explanations regarding LSG effectiveness in several countries; however, due to the limited number of cases, we did not make a chart/table.
Pag. 2/line 63: the authors pointed out that “...few studies have examined LSG development or its level of effectiveness in rural Asian contexts...”. In my opinion, it could be interesting to define the peculiarities of the Japanese context and explain why the LSG aspects of this country could be of interest for a larger audience and the scientific community.
Response: We have revised the description in the background section regarding the peculiarities of the Japanese context and the applicability to other settings.
Pag. 2/line 74 and ss.: Is unclear why the authors choose Unnan City as a case study. I appreciated a larger explanation of the reasons and the aspects that made this case interesting on the general portfolio of other cases. This focus makes the research more solid, otherwise, the selection could seem random or not enough motivated
Response: We have added a description of the choice of Unnan city as the first city that applied LSG to Japanese communities.
Pag. 3/line 96/97: the authors pointed out that “There are 30 multifunctional autonomies in Unnan City, each of which has various functions for managing their respective social issues”. I think it could be of interest to explain – and describe through data (population affected, area, density, the budget allocated for each autonomy) – the differences among different multifunctional autonomies, to point out how the identified LSG form is peculiar, interesting and fertile in the light developed by the authors.
Authors could have described better the context because I supposed that the effectiveness of the Self-Government depends on several factors, many of them connected to the local conditions and specific constraints. Provided data from annual city questionnaire are interesting, but not enough to understand the role of the LGS implemented in Unnan City as a good/effective practice because authors provide a weak framework for their study. As suggested, once the authors framed the research field and the two cases, the article is interesting and could become more solid.
Response: We have revised the description of autonomous community organizations. We have also added background information related to autonomous community organization and a figure of the location of Unnan city and districts.
Round 2
Reviewer 1 Report
The authors did a thorough revision according to reviewers comments. However the manuscript does not merit publication in its present form because of the way section 3 is presented.
The goal of having qualitative data is understanding the "why" instead of the "what". Interview quotes help the interpretation by showing how interviewees see the topic from their own perspective. These quotes should be intervowen with the author´s interpretation. In Section 3, however, the interpretation is given first, and then two quotes are included at the end. It is not clear what is it that the quotes show. And no further interpretation is made of them. This is not proper use of qualitative data nor is it quality science.
The authors should follow qualitative research guidelines for presenting evidence. These references may be a start: https://www.ncbi.nlm.nih.gov/pmc/articles/PMC6904374/
https://naepub.com/reporting-research/2020-30-3-2/
http://www.qualres.org/HomeDisp-3831.html
I believe the authors can do a much better job in this respect and wish them the best of luck in their revision.
Author Response
Point-by-Point Response
Please note that the changes made do not influence the content, conclusions, or framework of the paper. We have not listed below all minor changes made; however, these are highlighted in red font in the revised manuscript.
The authors did a thorough revision according to reviewers’ comments. However, the manuscript does not merit publication in its present form because of the way section 3 is presented.
The goal of having qualitative data is understanding the "why" instead of the "what". Interview quotes help the interpretation by showing how interviewees see the topic from their own perspective. These quotes should be interwoven with the author´s interpretation. In Section 3, however, the interpretation is given first, and then two quotes are included at the end. It is not clear what is it that the quotes show. And no further interpretation is made of them. This is not proper use of qualitative data nor is it quality science.
The authors should follow qualitative research guidelines for presenting evidence. These references may be a start: https://www.ncbi.nlm.nih.gov/pmc/articles/PMC6904374/
https://naepub.com/reporting-research/2020-30-3-2/
http://www.qualres.org/HomeDisp-3831.html
I believe the authors can do a much better job in this respect and wish them the best of luck in their revision.
Response:
We thank the reviewer for this insightful comment. We agree with the suggestion. Further, we read the suggested articles. As recommended, we have interwoven the quotes into our interpretation for better clarity. Accordingly, we have comprehensively revised the qualitative part of the results.
Reviewer 2 Report
In my opinion, the paper improved through the last version and I would thank the author for this effort.
Author Response
In my opinion, the paper improved through the last version and I would thank the author for this effort.
Response:
We thank the reviewer for their positive comment.